# Verification of the Field Productivity and Bioequivalence of a Medicinal Plant (*Polygonum multiflorum*) Developed Using an In Vitro Culture Method

**DOI:** 10.3390/plants9101280

**Published:** 2020-09-28

**Authors:** Yong-Goo Kim, Richard Komakech, Dae Hui Jeong, Kwonseok Jeon, Yunmi Park, Tae Kyoung Lee, Ki Hyun Kim, Byeong Cheol Moon, Youngmin Kang

**Affiliations:** 1Herbal Medicine Resources Research Center, Korea Institute of Oriental Medicine (KIOM), 111 Geonjae-ro, Naju-si 58245, Korea; kyg@kiom.re.kr (Y.-G.K.); bcmoon@kiom.re.kr (B.C.M.); 2Korean Medicine Life Science Major, Campus of KIOM, University of Science & Technology (UST), Daejeon 34054, Korea; richard@kiom.re.kr; 3Natural chemotherapeutics Research Institute (NCRI), Ministry of Health, Kampala P.O. Box 4864, Uganda; 4Forest Medicinal Resources Research Center, National Institute of Forest Science, Yeongju 36040, Korea; najdhda@korea.kr (D.H.J.); jks2029@korea.kr (K.J.); 5Department of Forest Bioresources, National Institute of Forest Science, Suwon 16631, Korea; pym5250@korea.kr; 6School of Pharmacy, Sungkyunkwan University, Suwon 16419, Korea; charmelon8@gmail.com (T.K.L.); khkim83@skku.edu (K.H.K.)

**Keywords:** *Polygonum multiflorum* Thunb., medicinal plant, field productivity, bioequivalence validation

## Abstract

*Polygonum multiflorum* Thunb. is a perennial plant that belongs to Polygonaceae. Root tissues are the main plant parts used as medicinal herbs in Korean oriental medicine. The *P. multiflorum* tuber is well known for its medicinal properties in Korean oriental medicine, and it contains a number of useful substances (secondary metabolites of emodin, 2,3,5,4′-tetrahydroxystilbene-2-*O*-*β*-d-glucoside (TSG), etc.) that are increasing in demand, as several studies show that they have beneficial effects on the human body. In this study, the production volumes and useful material content differences between cultured *P. multiflorum* seedlings (culture seedlings: CSs), which had been grown using a tissue culture technique under optimized conditions, and existing varieties in circulation (seed seedlings: SSs) were determined using a long-term field test. The growth characteristics of the underground parts were investigated by harvesting the tuberous roots (medicinal parts) after 1 year, and the results showed that the fresh and dry weights of the CS tubers were higher than those of the SS tubers. However, the SS rootlets had higher fresh and dry weights than the CS rootlets. A liquid chromatography-mass spectrometry component analysis of the *P. multiflorum* tubers and a Fourier transform near-infrared spectrophotometer analysis of the roots were undertaken. The results showed that the levels of TSG, which is a medicinal substance produced by *P. multiflorum*, were higher in the CSs than in the SSs, but the differences were not significant. The CS results from this study will inform future studies on the mass production of *P. multiflorum* in the field because the medicinal area was greater in CSs than in SSs.

## 1. Introduction

*Polygonum multiflorum* Thunb. is a tuberous root. It is shaped like a horn and is approximately 5–15 cm long and 3–10 cm in diameter [1]. *Polygonum multiflorum* is found in East Asia and North America and has long been used as a medicinal and ornamental plant [2]. It is also found in Henan, Guizhou, Sichuan, Jiangsu, and Hubei, China [3], where it has long been known as a medicinal plant.

In Korean herbal medicinal practice, *P. multiflorum* has been cleared to treat the liver, kidney, and blood and is used to treat hemorrhoids, etc. In addition, *P. multiflorum* has been widely utilized in traditional remedies because of its excellent blood pressure lowering effect and potential ability to reduce arteriosclerosis [4]. It has various other beneficial characteristics, such as cardio protective [5], nerve strengthening [6], antioxidant [7], anti-aging [8], anti-mutation [9], anti-arthritis [10], and skin improvement [11] properties. *Polygonum multiflorum* varieties are classified into two groups: red *P. multiflorum* and white *P. multiflorum*. Their roots commonly contain approximately 45% starch and 3% crude fat [12].

Phytochemicals are plant-derived chemicals, and some of these compounds help inhibit cell damage and improve immune function. To date, more than 10,000 species have been identified and they have varying efficacies depending on their type [13]. Phytochemical compounds are particularly abundant in edible and medicinal crops, such as vegetables and fruits. Representative examples include anthocyanins and carotenoids. *Polygonum multiflorum* contains the anthraquinone compounds chrysophanol, rhein, physcion, and emodin, as well as 2,3,5,4′-tetrahydroxystilbene-2-*O*-*β*-d-glucoside (TSG), a glycoside of anthraquinone compounds. TSG compounds are the largest group of phytochemical components in *P. multiflorum* [12,14]. Furthermore, antioxidative compounds can be derived from chrysophanol, rhein, physcion, emodin, and TSG. Apart from their antioxidative activities, these compounds have been shown to reduce cholesterol [4] and blood sugar [4,15] levels, have a whitening improvement effect by inhibiting melanin production [15,16], and reduce oxidative stress [17]. They also inhibit microorganisms that cause food poisoning, such as *Staphylococcus aureus* [18]. Therefore, *P. multiflorum* has various beneficial activities and is a potential high value-added medical food material.

*Polygonum multiflorum* is mainly proliferated using seed propagation and vegetative multiplication methods. Seed propagation requires a longer period of cultivation before the tuberous roots can be harvested than other breeding methods, and it is difficult to predict *P. multiflorum* growth in the following year because the germination rate varies depending on the nutritional status of the parent [19].

In addition, maintaining an elite line is problematic, because it is difficult to retain all of the superior properties of the parents during seed fruition. To compensate for these shortcomings, stem cutting nutrition methods are often used. However, it takes a long time to create the tuberous roots, and a cut seedling that is directly sown into the soil can easily be exposed to contaminants and damaged by infection. Contamination by viruses and other infectious agents reduces plant growth and decreases the quality of the tuberous roots, which leads to reduced herbal medicine production. Therefore, it is necessary to develop seedling cultivation techniques that can efficiently increase the quality of the tuberous roots and to establish a system for mass producing high-quality, uniform *P. multiflorum* seedlings. To solve this problem, our research team has succeeded in developing *P. multiflorum* tissue culture seedlings using an enhanced biomass through tissue culture techniques [20].

The aim of this study was to compare the production of root tubers, which have medicinal qualities, from the culture seedlings (CSs) and seed seedlings (SSs) of red and white *P. multiflorum* by field testing seedlings propagated using the optimized in vitro culture of *P. multiflorum* (CS) and comparing the results to those of SSs. This was a follow-up study to previous research on *P. multiflorum*. The results from this study will play an important role in improving the mass production of *P. multiflorum*.

## 2. Results

### 2.1. Comparison of the Plant Growth Characteristics of SS and CS-Derived Plants

The growth characteristics of the underground parts of the two *P. multiflorum* seedling types were investigated because the tuberous *P. multiflorum* root is used as a medicinal ingredient in Korean oriental medicine. The average fresh weight of all underground parts after *P. multiflorum* had been cultivated for 1 year was 327.9 ± 20.79 g for CSs and 243.5 ± 27.41 g for SSs. There was an 84.4 ± 34.40 g weight difference between the two types of *P. multiflorum* seedlings, and the total underground fresh weight was significantly higher for CS than for SS (Figure 1). The mean dry weights were 158.9 ± 13.0 g for CSs and 114.2 ± 14.62 g for SSs, which showed that the CS plants were heavier than the SS plants, although the difference was not significant (Figure 1). The mean fresh weights of the tubers alone were 291.1 ± 21.97 g for CSs and 167.0 ± 26.14 g for SSs. The CS tuber fresh and dry weights were significantly higher than the SS weights (Figure 1).

Conversely, the SS rootlet fresh and dry weights were higher than the CS rootlet weights. The area of the medicinal parts was measured using a root area machine. The results showed that the CS medicinal area was 83.76 cm^2^ and the SS area was 62.73 cm^2^ (Figure 2). The CS root tuber area was significantly larger than the SS tuber area (Figure 3). A visual assessment of *P. multiflorum* roots confirmed that the CS root biomass was relatively larger than the SS root biomass (Figure 3).

### 2.2. Multivariate Statistical Analysis and Component Analysis Using Fourier Transform Near-Infrared (FT-NIR) Spectrophotometry and Liquid Chromatography-Mass Spectrometry (LC-MS), Respectively

If the *P. multiflorum* roots from CS plants, which were developed using tissue culture techniques under optimized conditions (hormones, etc.), are to be used as raw materials for the preparation of drugs, then the active materials in the tubers need to be identified so that the differences between the existing SS-derived plants in circulation and the CS-derived plants developed using the new cultivation method can be determined. The FT-NIR results verified the bioequivalence test findings. The FT-NIR analysis showed the same pattern for the peaks representing the two types of seedlings (SS and CS) (Figure 4C). The SS and CS samples were subjected to LC-MS analysis for bioequivalence validation. This allowed us to compare the chemical compositions between SS and CS samples, including the main component, TSG (Figure 4B). Each sample was extracted with 70% MeOH/H_2_O using an ultrasonic cleaning bath. Then the chemical profile of the crude MeOH extract was tested using LC-MS. The LC-MS analysis revealed that the chemical profiles of the CS and SS extracts from *P. multiflorum* were almost identical (Figure 4D), and one main compound, TSG, was detected in the LC-MS UV chromatogram at both 254 and 315 nm. The similar chemical profile patterns indicated that comparable amounts of active compounds were produced by the CS- and SS-derived plants. A calibration curve was plotted using standard TSG so that the TSG concentrations in the samples could be reliably determined.

A range of five concentrations was used in triplicate to obtain a regression equation and to assess the response linearity (R^2^), limits of detection (LOD), and limits of quantification (LOQ). Linearity was evaluated by determining the R^2^ value of the equation derived from the peaks for the serially diluted five concentrations. The results showed that the LOD and LOQ were acceptably low (Table 1). The LC-MS analysis of the MeOH extracts from the two *P. multiflorum* types indicated that the quantitative levels of the TSG in the samples were 53.59 µg/mL for CS-derived plants and 50.75 µg/mL for SS-derived plants. The TSG levels were relatively higher in the CS-derived plants than in the SS-derived plants (Figure 4D).

## 3. Discussion

The optimal hormone concentration [20] for cultured *P. multiflorum* was identified using the results of a previous *P. multiflorum* study for optimizing in vitro culture conditions. The optimum media conditions derived from this previous study led to a large number of cultured *P. multiflorum* seedlings. Differences between the SS samples grown from the currently available products and the CS samples were determined in an experiment that compared the *P. multiflorum* growth characteristics over 1 year under field conditions. The underground *P. multiflorum* organs are the medically important parts. Therefore, a quantitative comparison of the underground and the aboveground biomasses was conducted. Most research on tissue culturing in Korea is related to the medicinal parts of Korean oriental medicine plants, which means that there has been previous research on root hypertrophy because it is mainly the roots that are used in traditional Korean medicine [21,22]. The results of the comparison between the SS and CS underground parts of *P. multiflorum* plants showed that the underground biomass of the CS samples was higher than that of the SS samples. However, the SS small lateral roots had large fresh weights. Plant roots adapt to changes by adjusting their shape and area according to gravity, light, and soil conditions [23]. Furthermore, plant growth hormones play important roles in root growth and architecture [24,25]. In particular, the auxin group of plant hormones (phytohormones) is closely involved in root development. In contrast, cytokinin hormones are known to inhibit lateral root development [26]. It is thought that *P. multiflorum* CS plants are greatly influenced by phytohormones. The differences in biomass, tuber root, and lateral root development between the CS and SS samples were probably because of the auxin and cytokinin hormones as their levels were higher under the optimal culture conditions than would be expected under field conditions. The plant hormone gibberellic acid, which is associated with the formation of tuber roots, has been extensively studied [27]. Transcriptome based RNA-Seq analyses have been used in hormone studies to investigate the tuber root development process and its relationship to changes in plant hormone biosynthesis. The role of sucrose and glucose in plant physiology is related to seed germination and the control of blooming. In particular, sucrose is closely related to the development of food crop tubers [28]. Furthermore, the production of medicinal resources is related to the formation of tuber roots in *Rehmannia glutinosa* and *Pinellia ternata* [21,22]. Therefore, the formation of *P. multiflorum* tubers may also be partly owing to sucrose, and previous research on Korean oriental medicine plants [20] has suggested that this may explain why CS-derived plants have higher fresh weights than SS-derived plants. Multiplication of the tubers can be visually assessed from the shape of the CS tubers (Figure 3). This lack of difference in the underground biomasses could be because of a genetic variation in *P. multiflorum*. However, an important point about this study is that the CS-derived plants were developed using an optimized tissue culture system and that this probably led to the underground parts developing more rapidly than those in the conventional SS-derived plants. This results from this study suggested that *P. multiflorum* could be harvested, which would reduce labor and production costs because it takes less time to produce a viable crop. Shortening the growth period or increasing biomass is often considered to be very important aspects of crop cultivation [29].

The FT-NIR pattern confirmed that there were no differences between the CS and SS samples and that the starch and crude fat content in CS- and SS-derived plants was almost the same. This confirmed CS stability and meant that CS material could potentially be used in herbal medicines. If the CS material is to be developed further, then it is essential to determine the gene distribution, expression, and genotype differences between the CS and SS samples to confirm that the CS-derived plants are stable as a food or medicine. The FT-NIR experimental results suggested that there were no significant differences between the two types. However, more accurate analyses that include additional statistical tests based on the FT-NIR patterns should be undertaken. The LC-MS results showed that the TSG concentration (the *P. multiflorum* index material) was more than 3% higher in the CS samples than in the SS samples, which confirmed that CS-derived plants showed other improvements over SS-derived plants apart from biomass. An analysis of emodin-8-*O*-*β*-d-glucoside, physcion-8-*O*-*β*-d-glucoside, emodin, and physcion compounds will also be conducted after this study. In addition, the stability evaluation of TSG for use as a drug should be further improved based on the study results [30]. The effects of metal ions on the stability of the TSG used as a standard material in this study should also be investigated.

## 4. Materials and Methods

### 4.1. Plant Materials, Transplanting, and Harvesting

This experiment aimed at validating the productivity of *P. multiflorum,* which is a vital herbal medicine resource plant in the field. The experiments were conducted at the National Forest Medicinal Resources Research Institute located in Punggi-eup, Yeongju, Gyeongsangbuk-do, South Korea. Two types of seedlings, namely, SS and CS, were grown under suitable conditions for field transplantation. They were then transplanted into soil and grown for forty (40) weeks (Figure 5A,B). The annual average temperature and humidity of the planting season were not different from those of the previous year and there were no natural disasters, such as cold or wet weather. The transplanted soil area was divided into three 6 × 6 m plots, and the two types of seedlings were randomly transplanted at 15 × 15 cm intervals from left to right in each area. A total of 40 transplanted plants of each type were planted in one plot. The transplanted seedlings were watered on the first day after transplantation to enable them to adapt to the soil. However, no further water, fertilizer, or pesticides were applied.

The *P. multiflorum* plants were harvested in November of the same year and its production levels and growth conditions were determined for the SS- and CS-derived plants (Figure 5C).

### 4.2. Assessment of the Underground Polygonum Multiflorum Parts

The fresh weight of the harvested roots tubers were obtained and their lengths measured using a ruler. The tubers were then dried in an oven at 150 °C for 3 days. The exact root surface area was measured using WinRHIZO (Regent Instruments, Quebec, QC, Canada) and three parameters of projected area (PA), surface area (SA), and volume (Vol) were determined. The roots were spread on a root-positioning tray and scanned using a flatbed scanner (GT-20000, EPSON, Nagano, Japan). The acquired images were analyzed using WinRHIZO Pro software, v.2005a (Regent Instruments, Quebec, QC, Canada). Finally, the overall fresh and dry weights of all underground parts were measured and compared.

### 4.3. FT-NIR Analysis and Quality Inspection

FT-NIR spectra were recorded on a TANGO FT-NIR spectrometer (BRUKER Corporation, Billerica, Massachusetts, USA) equipped with an NIR fiber optic probe (type 847-072200), interferometer, InGaAs detector and broadband light source (50 W), and a quartz halogen lamp that provides interaction measurements. A sample of *P. multiflorum* was placed in a holder with the stem-axillary axis horizontal. The interaction spectrum for each sample was measured using three opposite equatorial positions and the average spectrum for each sample was analyzed. The source and detector fibers were randomly placed on top of the branch cable. Light was directed to the sample using a source fiber, separated from the sample by a detector fiber and directed to a TANGO FT-NIR spectrometer with a spectral range of 800–2500 nm. The mirror speed was 0.9494 cm/s and the resolution was 16/cm. Place the branched optical probe at an angle of 75° to the level to avoid surface reflection and guaranteed subsurface penetration of light into *P multiflorum* samples.

### 4.4. Extraction

Extract samples (10 g each from CSs and SSs) were prepared for LC-MS analysis. Each sample was ground and extracted with 70% MeOH/H_2_O (250 mL) twice over 90 min using an ultrasonic cleaning bath (Model 8510E—DTH, Branson, USA). The resultant solution was then filtered using Whatman filter No. 1 (Whatman, Maidstone, U.K.) and concentrated under vacuum reduced pressure at 40 °C and 55 rpm using an EYELA N-1200B rotary evaporator (Tokyo Rikakikai Co. Ltd., Japan) to obtain the MeOH extract (1.2 g), which was then used in the LC-MS analysis. The same extraction procedure was conducted in triplicate.

### 4.5. LC-MS Analysis for Bioequivalence Validation

LC-MS analysis was carried out using the Agilent 1200 Series analytical system (Agilent, Santa Clara, CA, USA) equipped with a photodiode array detector integrated with a 6130 Series electrospray ionization mass spectrometer. Each crude extract from the samples (1.0 mg) was first dissolved in 50% aqueous methanol (1.0 mL). Methanol was then added again to obtain a total solution volume of 100 μg/mL before filtering with a 0.45 mm hydrophobic polytetrafluoroethylene (PTFE) filter. The resultant filtrate was then subjected to LC-MS analysis with a Kinetex C_18_ column (2.1 × 100 mm, 5 μm; Phenomenex, Torrance, CA, USA) at 25 °C. Formic acid in water [0.1% (*v*/*v*)] (A) and methanol (B) were used as the mobile phase, which was delivered at a flow rate of 0.3 mL/min based on a programmed gradient elution of 10–100% (B) for a duration of 30 min, followed by 100% (B) for 1 min, 100% (B) isocratic for 10 min, and 10% (B) isocratic for 10 min to perform post-run column reconditioning.

### 4.6. Quantitative Analysis of TSG

The TSG was analyzed using LC-MS (Agilent, Santa Clara, CA, USA). A calibration curve and a linear regression equation were generated for the external TSG standard. The standard TSG stock solution was prepared in MeOH at 1.0 mg/mL, and work solutions were mixtures of the stock solution after series dilutions with methanol to obtain five concentration levels. The work solutions were filtered through a 0.45 mm hydrophobic PTFE filter prior to LC-MS injection. The linearity was plotted after undertaking a linear regression analysis of the integrated peak areas (Y) vs the concentration of each standard (X, µg/mL) at the five different concentrations. Each *P. multiflorum* crude MeOH extract (1.0 mg) was dissolved in MeOH (1.0 mL), which generated the sample stock solution. This was further diluted with MeOH to provide a 100 µg/mL solution. The solution was filtered through a 0.45 mm hydrophobic PTFE filter and finally subjected to LC-MS analysis with a Kinetex C_18_ column (2.1 × 100 mm, 5 µm; Phenomenex, Torrance, CA, USA) set at 25 °C. The mobile phase consisted of formic acid in water [0.1% (*v*/*v*)] (A) and methanol (B) and was delivered at a flow rate of 0.3 mL/min, and the following programmed gradient elution was applied: 10–100% (B) for 30 min, 100% (B) for 1 min, 100% (B) isocratic for 10 min, and 10% (B) isocratic for 10 min to perform post-run reconditioning of the column. The injection volume was 10 µL. The quantification of TSG was based on the peak area under the line obtained after MS detection in selected ion monitoring mode and calculated as equivalents of the standard. The content was expressed as µg per 1 mg of extract weight. The sensitivity was assessed by determining the LOD and LOQ, which were determined by the signal-to-noise (S/N) method. The S/N ratios were 3 and 10 for LOD and LOQ, respectively.

### 4.7. Data Recording and Statistical Analysis

The data were subjected to analysis of variance and mean separation at a significance level of *p* = 0.05. The data were also subjected to unpaired t-tests with Welch’s correction. GraphPad Prism (GraphPad software, Ver. 5. 03) and R statistics were used for the analyses.

## 5. Conclusions

When the CS plants, developed using plant tissue cultivation under optimized conditions, were grown in the field, the results showed that the medicinally active area of the underground parts was larger than that of the SS plants and that the CS-derived plants produced more active material than the SS-derived plants. The findings showed that it is possible to efficiently mass produce *P. multiflorum* for medicinal purposes. The results also suggested that it should be possible to reduce labor requirements and production times and costs. Furthermore, we showed that the *P. multiflorum* medicinal areas reached a state where they could be harvested more rapidly using tissue culturing than was possible using conventional growing methods.

## Figures and Tables

**Figure 1 plants-09-01280-f001:**
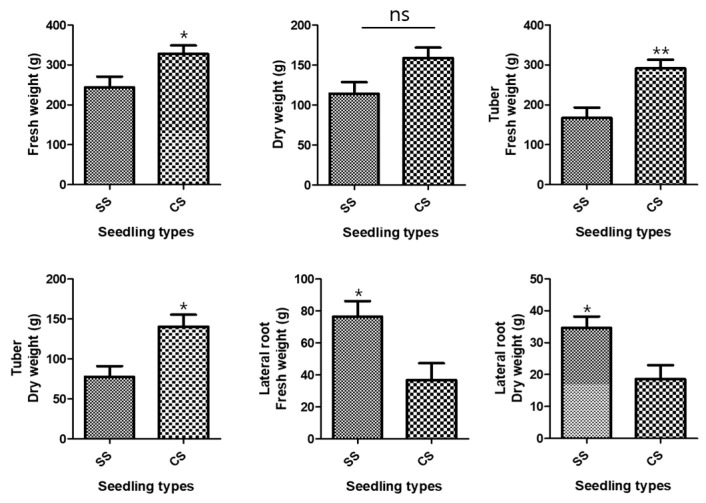
Development of the underground parts. Comparisons between the two types of *Polygonum multiflorum* seedlings [seed seedling (SS) and culture seedling (CS)]. *n* = 10 samples containing all the underground parts; bar graphs represent each treatment mean ± standard error of the mean (SEM). Unpaired t-test with Welch’s correction was used to determine significant differences between SS and CS (*ns* = not significant, * *p* < 0.05, ** *p* < 0.01).

**Figure 2 plants-09-01280-f002:**
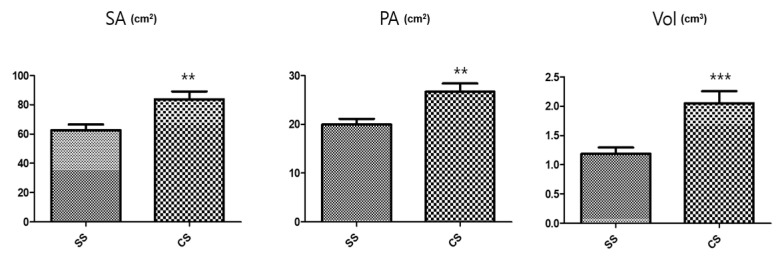
Size characteristic comparisons of the medicinal root tubers in the different seedling types. SA: Surface area; PA: Projected area; and Vol: Volume. *n* = 5 root tubers; bar graphs represent each treatment mean ± standard error of the mean (SEM). Unpaired *t*-test with Welch’s correction was used to determine significant differences between the seed seedling (SS) and culture seedling (CS) types (** *p* < 0.01, *** *p* < 0.001).

**Figure 3 plants-09-01280-f003:**
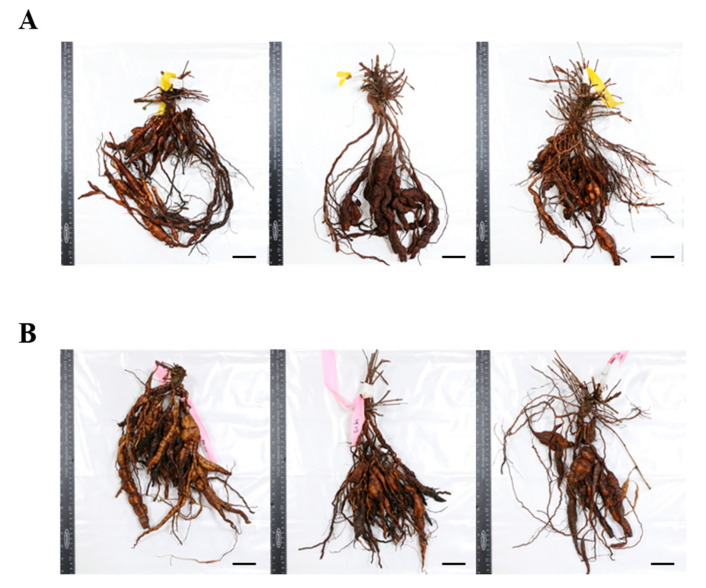
Comparison of the 1st year seedling underground root morphologies. (**A**) Seed seedling (SS) and (**B**) culture seedling (CS). Scale bar indicates 5.0 cm.

**Figure 4 plants-09-01280-f004:**
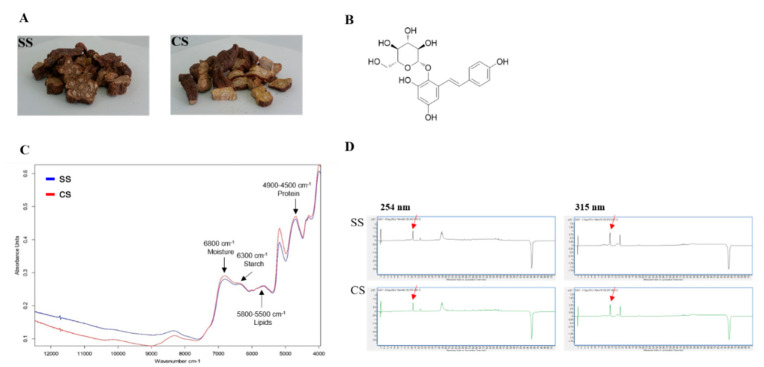
Liquid chromatography-mass spectrometry (LC-MS) analysis of the two types of *Polygonum multiflorum*: (**A**) seed seedling (SS) and culture seedling (CS). (**B**) Chemical structure of 2,3,5,4′-tetrahydroxystilbene-2-*O*-*β*-d-glucoside (TSG). (**C**) Fourier transform near-infrared spectrum (Blue peak SS; Red peak CS). (**D**) UV chromatogram of the LC-MS results for the CS- and SS-derived plant. The detection wavelengths were 254 and 315 nm, and the red arrow indicates the TSG peak.

**Figure 5 plants-09-01280-f005:**
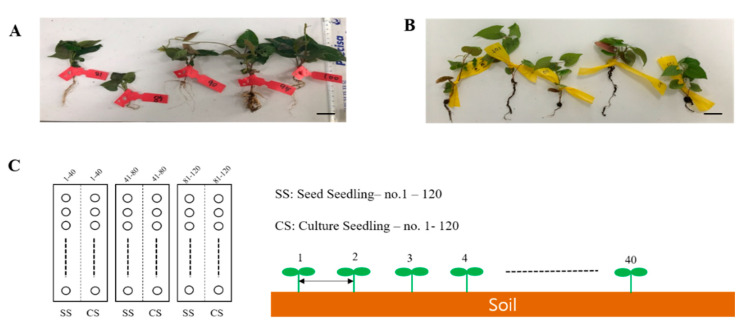
Field experiment seed plot design for the two types of *Polygonum multiflorum* seedlings. (**A**) SS, seed seedling. (**B**) CS, culture seedling. Scale bar indicates 1.0 cm. (**C**) Planting interval and number of *P. multiflorum* planted.

**Table 1 plants-09-01280-t001:** Linearity of standard curve and detection/quantification limits for the standard (2,3,5,4′-*O*-*β*-d-glucoside) and its quantitative analysis in the two samples of *P. multiflorum*.

Compound	2,3,5,4′-Tetrahydroxystilbene-2-*O*-*β*-d-glucoside (TSG)	Samples	Contents of TSG (μg/mg) ^c^
R^2 a^ LOD (μg/mg) ^b^	0.997 0.073	SS CS	50.75 (5.075%) 53.59 (5.359%)

Ay, peak area; x, concentration of the standard (μg/mL); ^a^ R^2^, correlation coefficient for 5 data points in the calibration curves (*n* = 3); ^b^ LOQ, limit of quantification (S/N = 10); ^c^ Mean ± SD (*n* = 3).

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
