# Peer review of "Verification of the Field Productivity and Bioequivalence of a Medicinal Plant (Polygonum multiflorum) Developed Using an In Vitro Culture Method"

_plants, 2020, doi:10.3390/plants9101280_

Round 1
Reviewer 1 Report
I have concluded the review of the manuscript plants-925358 titled as «Verification of the Field Productivity and Bioequivalence of a Medicinal Plant (Polygonum multiflorum) Developed using an In Vitro Culture Method»
In this work, the authors provided an interesting approach in improving the mass production of P. multiflorum root tubers, a valuable source for the natural product 2,3,5,4’-tetrahydroxystilbene-2-O-β-D-glucoside (TSG). My observations focused mainly on the area of my specialty which is the phytochemical part of the text.
In my point of view, the manuscript presents several parts that need to be improved, omissions as well as parts that need to be clarified and reworded more clearly. The whole analytical part concerning the quantification of the main component TSG is difficult to comprehend. The use of infrared spectroscopy is also vague and difficult to understand, while the conclusions that emerge are relatively arbitrary.
Specifically:
Line 26. Greek letters in nomenclature should be given in Italics form
Line 32. Year not years
Lines 91-92. This phrase highlights a conclusion while the paragraph presents the aim of this work.
Sub-section 2.1. Lines 94-103. I believe that ±SD values should be given in the text alongside to mean root weights.
Sub-section 2.2. Analysis of the FT-NIR (curves assignment) is completely absent.
Lines 131-132. The sentences are vague. What do you mean by “bioequivalence test” and “drug equivalence validation”. Although this manuscript isn’t a full paper manuscript it should so difficult to comprehend.
Line 133. In the text, there are pieces of information concerning the quantification of only one phytochemical ingredient, meaning the TSG. So, what chemical profile?
Lines 136-138 and Fig 4.D. In general, MeOH is considered as a “universal solvent”, meaning that when it is used as a pure solvent or in this case as an aquatic (?) mixture (authors does not clarify it) to extract phytochemicals from a plant material the crude isolate is usually comprising from a broad chemical range of ingredients. Surprisingly, in this case, the LC chromatograms are quite “clear”, as if the crude extract was pre-treated. Please, do discuss this.
Lines 203-204. Strongly disagree. There are no data given in the text or discussion that supporting the current conclusion.
Lines 201-213. From which part of the experimentation this conclusion comes from?
Lines 213-217. These are future aspects and not experimental conclusions. It should be in the discussion section.
Sub-section 4.4. It should be re-written. Crucial data are missing e.g. volume of the methanolic solution, the number of extractions per sample to ensure complete isolation of TSG, replicates of the extraction procedure, concentration to dryness, etc.
Author Response
We attached the file below. Thank you for your cooperation.

Reviewer 2 Report
Your manuscript deals with the production volumes and biological active molecules content differences between cultured P. Multiflorum seedlings and existing varieties in circulation. These differences were determined using a long term field test. In particular, your study focused on the comparison of the production of root tubers, endowed with medicinal qualities, from the culture seedlings and seed seedlings.
The scientific procedures of your study are clearly described and the aim of your work is well supported by the final results. The results also suggested that it should be possible to reduce labor requirements and production times and costs.
For these reasons, your manuscript is certainly worth being published as communication, in present form.
Author Response

(The authors gave the same response as above.)

Round 2
Reviewer 1 Report
In the present form, it could be considered for publication.